# Horizontal Bi-Stable Vibration Energy Harvesting Using Electromagnetic Induction and Power Generation Efficiency Improvement via Stochastic Resonance

Linshi Guo [1], Wei Zhao [2], Jingchao Guan [1], Nobuyuki Gomi [1] and Xilu Zhao [1,*]

1   Department of Mechanical Engineering, Saitama Institute of Technology, Saitama 369-0293, Japan
2   Weichai Global Axis Technology Co., Ltd., Tokyo 107-0062, Japan
*   Correspondence: zhaoxilu@sit.ac.jp

**Abstract:** In this study, a vibration energy-harvesting system is developed by first proposing a horizontal bi-stable vibration model comprising an elastic spring and a mass block and then applying an electromagnetic induction power generation device composed of a magnet and a coil. Subsequently, based on a weight function that considers the mutual positional relationship between the magnet and conducting coil, a set of simultaneous governing equations that consider the elastic force of the elastic spring and the Lorentz force of electromagnetic induction is derived. Additionally, a numerical analysis method employing the Runge–Kutta method is utilized to obtain a numerical solution for the vibration response displacement and vibration power generation voltage simultaneously. Experiments are performed to verify the results yielded by the proposed bi-stable vibration energy-harvesting system. The results shows that the measured vibration response displacement and the vibration power generation voltage are consistent with the analytical results. Moreover, issues including the identification of damping coefficients that consider the mutual effects of normal kinetic friction and electromagnetic induction damping forces, as well as the effects of electromagnetic induction damping on the vibration response displacement, are discussed comprehensively. Simultaneously adding random and periodic signals to the bi-stable vibration model results in stochastic resonance and improves both the vibration amplification effect and vibration power generation.

**Keywords:** vibration energy harvesting; bi-stable vibration system; stochastic resonance; nonlinear vibration system; vibration power generation

## 1. Introduction

To further develop natural renewable energy sources, including the ocean and wind, studies pertaining to vibration energy harvesting have received considerable attention, and numerous findings have been reported [1–4]. Vibration energy harvesting is a power generation method in which vibrational energy can be converted into electrical energy. The development of vibration devices with large amplitudes is desirable as greater vibration amplitudes result in higher amounts of vibration power generated.

Previously, vibration energy harvesting was conducted using amplitudes amplified via resonance, where the natural frequency of the vibration system had to match the frequency of the environmental vibration [5–7]. However, the disadvantage of the linear resonance system is that large amplitudes cannot be stably continued due to complex frequency components in natural environment. Hence, a method of installing multiple vibration models featuring different natural frequencies in a single vibration system has been proposed; however, it can be used for only a limited range of applications and requires a larger device [8,9]. Therefore, nonlinear vibration systems have been suggested for vibration environments featuring multiple frequency components; however, their low vibration amplitudes hamper their practical application in vibration energy harvesting [10–12].

Hence, new research and development programs for vibration energy harvesting have been proposed that exploit the advantages of stochastic resonance, which can magnify the vibration amplitude of a bi-stable vibration model considerably in a random noise environment [13,14]. Stochastic resonance is a physical phenomenon in which the response signal may be amplified significantly at a specified probability via the addition of a weak periodic signal to a nonlinear system subjected to a random excitation signal. In fact, numerous theoretical discussions for elucidating this phenomenon have been published [15–19].

In the field of mechanical systems, for bi-stable vibration systems subjected to random excitation signals, use of a stochastic resonance phenomenon, in which the response vibration of the system may be amplified significantly by adding a periodic input signal, has been applied for vibration power generation [20–23]. Therefore, bi-stable vibration systems that allow stochastic resonance to occur in mechanical systems must be developed.

Currently, the most frequently investigated bi-stable vibration systems for mechanical systems are those that employ cantilevered beams with mass blocks mounted at their tips [24–26]. Two approaches are typically employed to achieve bi-stable vibrational characteristics. The first uses the opposing repulsive forces between permanent magnets placed at the tip of a cantilever beam and at a short distance from it [27–30]. The other uses the elastic force caused by the gravity of the mass block attached to the tip of an inverted cantilever, which causes the cantilever to bend in the lateral direction [31–33]. In addition, to improve the bi-stable vibration performance, a number of bi-stable vibration models have been investigated, including a bi-stable vibration model [34–36] that combines multiple cantilever beams, a bi-stable vibration model featuring a modified cantilever beam shape [37–39], a bi-stable vibration model comprising two end-support beams bent to the side via the application of a compressive load in the axial direction [40–43], and a tri-stable vibration model [44–49] comprising a modified number of magnets to be installed at and near the tip of a cantilever beam.

In practical applications of vibration energy harvesting, bi-stable vibration models of mechanical systems in natural vibration environments often oscillate at relatively low amplitudes induced by random excitation signals. Utilizing significantly amplified bi-stable vibrations by intentionally introducing periodic excitation signals to a bi-stable vibration model to achieve stochastic resonance is a promising method.

To achieve the abovementioned objective, Zheng et al. developed a bi-stable vibration model comprising a horizontally mounted cantilever beam and a magnet attached to the tip of the beam. By inputting random noise and an external periodic force in the vertical direction, stochastic resonance was re-created in the bi-stable model and the findings were discussed [50]. Nakano et al. investigated the conditions under which the input vibration of a cantilever beam is modified in the horizontal direction and stochastic resonance occurs [51]. In addition, Zhang et al. evaluated the utility of a vibration energy-harvesting system for power generation equipment with bi-static vibrations, which is appropriate for automotive tire applications [52,53].

Nevertheless, a stable vibration limit exists when cantilevered beams are used. Moreover, the cantilevered beam structure cannot accommodate accidental disturbance loads, which suggests the significance of large-scale vibration models that allow for steadier long-term vibrations [54].

Therefore, Zhao et al. suggested a large-scale bi-stable vibration model that comprises a spring sustained from an oblique direction and a mass block for developing a bi-stable vibration system that can stably produce stochastic resonance while ensuring an exceptionally large amplitude, which is better than the conventional bi-stable vibration systems [55].

Methods for converting vibration energy into electric power, including piezoelectric methods based on piezoelectric and electromagnetic induction methods based on magnets and coils, have been considered for vibration power generation [56]. The electromagnetic induction method has become a typical method for generating vibrations.

Herein, a vibration energy-harvesting system is developed by first proposing a horizontal bi-stable vibration model comprising an elastic spring and a mass block and then applying an electromagnetic induction power generation device composed of a magnet and a coil. By proposing a weight function that considers the mutual positional relationship between the magnet and coil during bi-stable vibration motions, a simultaneous set of governing equations that simultaneously considers the elastic force of the spring and the Lorentz force of electromagnetic induction was derived. By applying the Runge–Kutta method to the simultaneous equations, numerical solutions for the vibration response displacement and vibration generation voltage of the mass block were investigated simultaneously. Furthermore, the identification of damping coefficients while considering the mutual effects of kinetic friction and electromagnetic induction damping forces and the effects of electromagnetic induction damping on the vibration response displacement are discussed comprehensively. When random and periodic excitation signals are applied to a large-scale bi-stable vibration model, stochastic resonance is generated, which allows the effect of vibration amplification and the amount of electricity generated via vibration power to be determined.

## 2. Material and Method

### 2.1. Horizontal Bi-Stable Vibration Harvesting System

A horizontal bi-stable vibration energy-harvesting model, as illustrated in Figure 1, is proposed herein. The mass block with rotating rolls on the horizontal rail is allowed to propagate in the left and right directions. Similar elastic springs are joined to each side of the mass block by a rotating pin, and a guide rink is attached to the center of the elastic spring. Two permanent magnets are placed parallel to each other in the mass block, and several conductive coils are arranged in the central space between the two permanent magnets. An induced voltage can be generated across the coils when a relative motion exists between the magnets and coils. The experimental apparatus is placed in a mini-shaker, which is then placed in a general shaker. Therefore, the mini- and general shakers can be vibrated simultaneously.

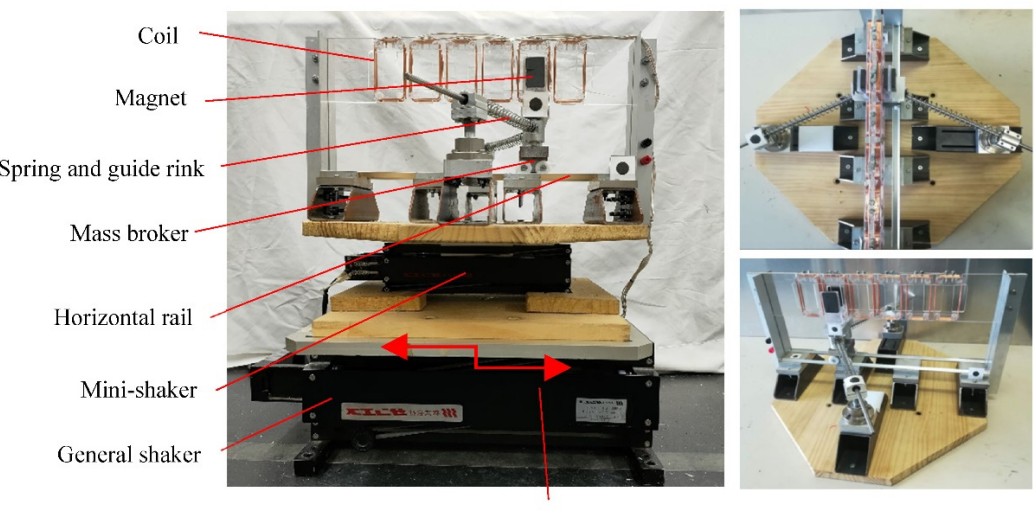

**Figure 1.** Horizontal large-scale bi-stable vibration-harvesting experimental equipment.

The results of the experimental results are measured using a high-speed camera to capture images of the mass block and the measurement markers affixed to the support stand (see Figure 2). Subsequently, the resulting video data are loaded into the computer, and tracking software is used to create a time series of the vibration displacement data. A data logger is used to record the vibration-generated voltage in the coil. A photograph of the experimental apparatus is shown in Figure 3, and the specifications of its components are listed in Table 1.

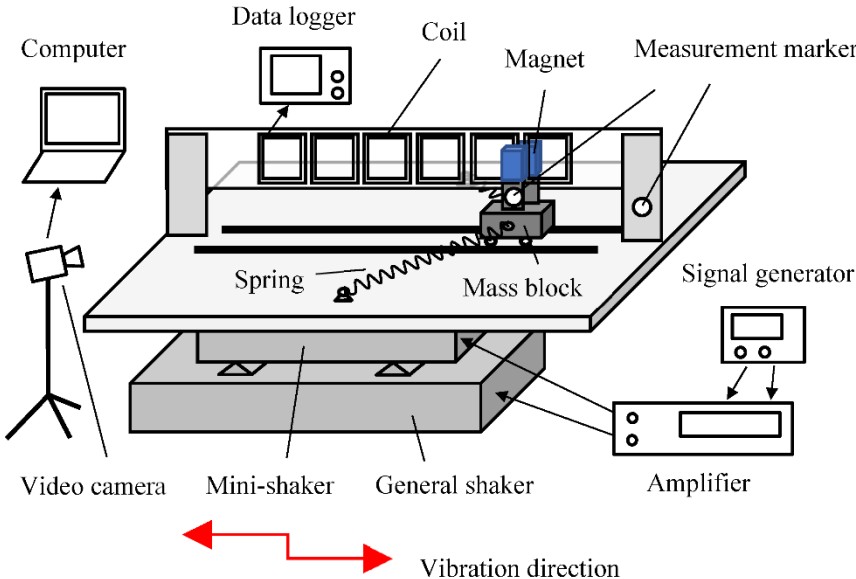

**Figure 2.** Measurement system for bi-stable vibration harvesting.

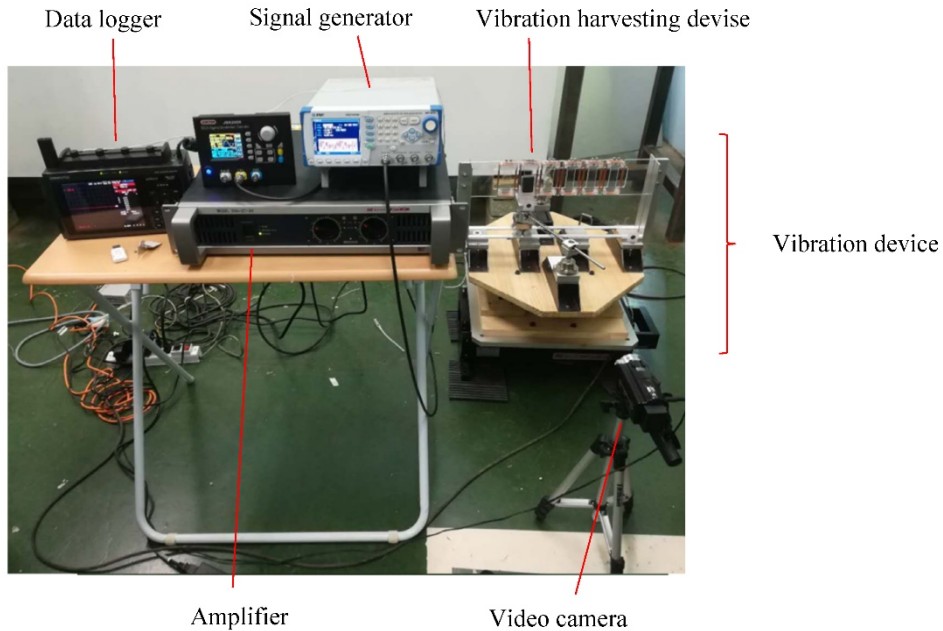

**Figure 3.** Experimental equipment for verification of bi-stable vibration harvesting system.

The main vibration unit of the motion model is simplified, as depicted in Figure 4a, where $m$ represents the mass of the mass block, $x_d$ the displacement of the mass block from the axis of symmetry, $x_t$ the displacement of the support point from the axis of symmetry, $K$ the spring coefficient, $F$ the elastic force of the spring, $F_c$ the damping force resulting from the motion friction, $F_L$ the Lorentz force yielded by the coil current in the magnetic field, and $h$ the vertical distance from the support point to the center point of the mass block, $\theta$ the angle between the axis of the spring and the horizontal direction. However, because the initial length $l_0$ of the elastic spring is larger than the distance $h$, two static equilibrium positions exist on the mass block, i.e., one on each side of the axis of symmetry, thus yielding three states of vibration, as indicated in Figure 4b–d. Figure 4c,d show monostable vibrations, whereas Figure 4b show as bi-stable vibration.

**Table 1.** Settings of experimental system.

| Item | Detail | Parameter |
|---|---|---|
| Mass block | weight | 910 g |
| Horizontal rail | Length | 350 mm |
| | Width | 50 mm |
| Vertical distance | From rail to support base | 150 mm |
| Elastic spring | Spring coefficient | 157 N/m |
| | Initial length | 180 mm |
| Permanent magnet | Length | 40 mm |
| | Width | 30 mm |
| | Thickness | 15 mm |
| | Surface magnetic flux density | 80 mT |
| Conductor coil | Frame width | 50 mm |
| | Frame height | 85 mm |
| | number of coil turns | 150 |
| | Electric resistance | 30 Ω |
| Mini-Shaker | SSV-105 | SAN ESU Co., Ltd. |
| General Shaker | SSV-125 | SAN ESU Co., Ltd. |
| Amplifier | SVA-ST-30, two channels | SAN ESU Co., Ltd. |
| Video recording information | Video camera | GZ-E765, JVC Co., Ltd. |
| | Frames per second (FPS) | 300 |
| | Dot per inch (DPI) | 1920 × 1080 |
| | Diameter of the marker | 10 mm |
| General function Generator | NF-WF1973 | NF Corporation |
| Mini-function Generator | JDS2800 | Hangzhou Measurement Instrumentation Co. |
| Data logger | GL2000 | Graphtec Co. |
| Marker tracking software | MOVIS Neo V3.0 | NAC Image Technology Inc. |

Based on a bi-stable vibration system that vibrates monostatically within practical random vibration environments, the key point of this study is to exploit the characteristics of the large amplification effect yielded the moment when the mass block is subjected to periodic excitations and transcends the potential energy peak in the center of the system.

Based on Figure 4a, the equation of motion of the mass block along the *x*-direction is formulated as follows:

$$m\ddot{x}_d + c(\dot{x}_d - \dot{x}_t) + F_L + 2F\cos\theta = 0 \tag{1}$$

where *c* is the damping coefficient. Therefore, the elastic force *F* and angle *θ* can be estimated as follows:

$$F = K\left(\sqrt{(x_d - x_t)^2 + h^2} - l_0\right) \tag{2}$$

$$\cos\theta = \frac{x_d - x_t}{\sqrt{(x_d - x_t)^2 + h^2}} \tag{3}$$

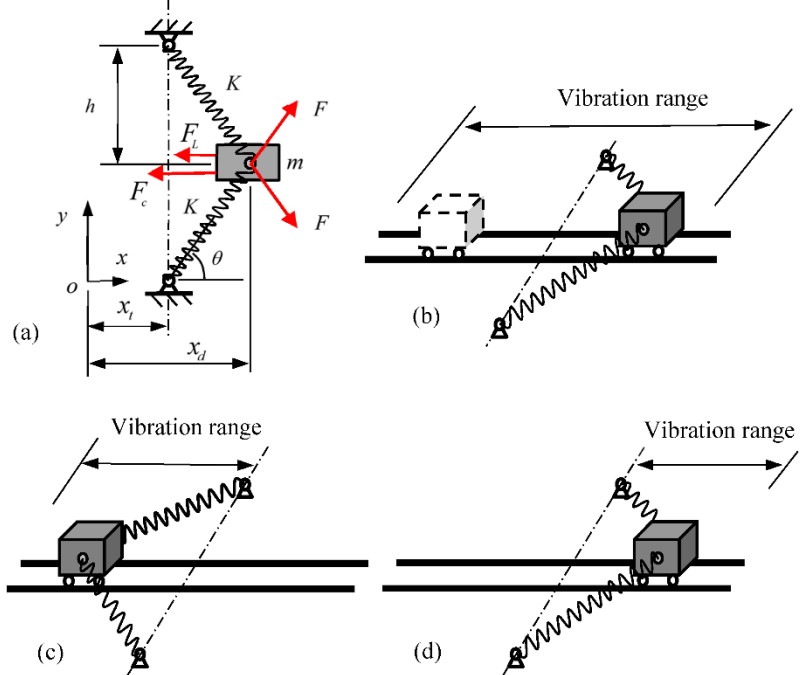

**Figure 4.** Analytical model and bi-stable vibration: (**a**) Bi-stable vibration model, (**b**) monostable vibration on right side, (**c**) monostable vibration on right side, and (**d**) bi-stable vibration.

Here, $K$ is the spring constant. The equation of vibration is obtained by substituting Equations (2) and (3) into Equation (1), and the resulting equation is as follows:

$$m\ddot{x}_d + c\left(\dot{x}_d - \dot{x}_t\right) + F_L + 2K\left(1 - \frac{l_0}{\sqrt{\left(x_d - x_t\right)^2 + h^2}}\right)(x_d - x_t) = 0 \tag{4}$$

The relative displacement between the mass block and support point is expressed as follows:

$$x = x_d - x_t \tag{5}$$

The equation of vibration for a relative displacement $x$ is expressed as follows after substituting Equation (5) into Equation (4).

$$m\ddot{x} + c\dot{x} + F_L + 2K\left(1 - \frac{l_0}{\sqrt{x^2 + h^2}}\right)x = -m\ddot{x}_t \tag{6}$$

The magnet and coil sections are illustrated in a simplified form in Figure 5, where $a$ and $b$ are the width and height of the magnets, respectively. Six sets of coils are mounted in a row with the same spacing between the magnets, and the terminals between the coils are joined, as illustrated in Figure 5. The width $L_c$ of the coils is wider than the width $a$ of the magnets. If the magnets are considered in vibration, as depicted by the blue arrows, then a current $I$ is yielded in the direction of the red arrow as the magnets transit the front of the coils, and the voltage magnitude thereof can be estimated as follows:

$$V_{\max} = Bnb\dot{x} \tag{7}$$

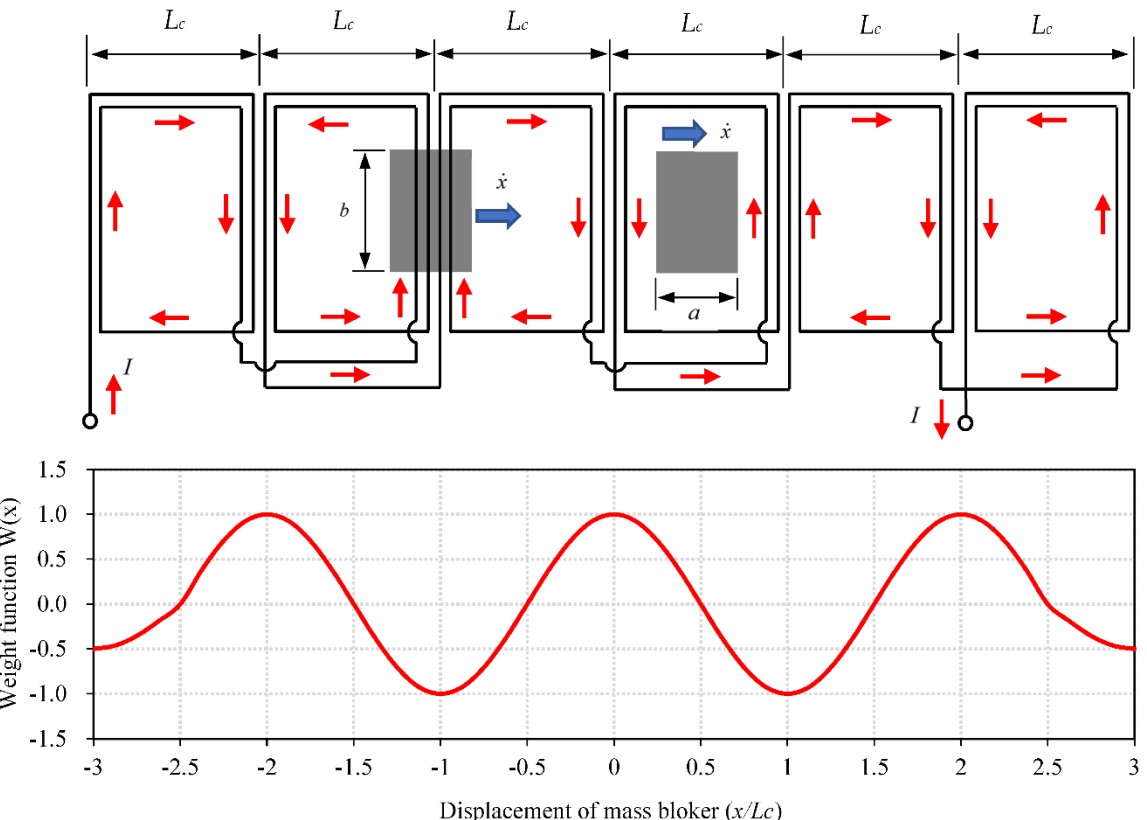

**Figure 5.** Magnet and coil positioning and placement weight function $W(x)$ for vibratory power generation.

Here, $n$ represents the number of coils turns, and $\dot{x}$ the velocity of the magnet vibration. Because the voltage decreases to zero when the magnet transits the center of the coil, the placement weight function $W(x)$ illustrated in Figure 5 is expressed as follows:

$$W(x) = \begin{cases} \sin \frac{\pi}{L_c} x & -2L_c \leq x \leq 2L_c \\ \frac{1}{2} \sin \frac{\pi}{L_c} x & other \end{cases} \tag{8}$$

Substituting the placement weight function $W(x)$ in Equation (8), the electromagnetic induction voltage $V$ yielded in the coil is expressed as follows:

$$V = BnbW(x)\dot{x} \tag{9}$$

The Lorentz force $F_L$ generated against the mass block by the current in a coil in the magnetic field is expressed as follows:

$$F_L = \frac{V}{R}Bnb \tag{10}$$

where $R$ represents the electrical resistance of the coil. Subsequently, substituting Equation (9) into Equation (10) yields:

$$F_L = \frac{B^2 n^2 b^2}{R}W(x)\dot{x} \tag{11}$$

Next, substituting Equation (11) into Equation (6) yields the following equation of motion:

$$m\ddot{x} + \left[ c + \frac{B^2 n^2 b^2}{R}W(x) \right] \dot{x} + 2K \left( 1 - \frac{l_0}{\sqrt{x^2 + h^2}} \right) x = -m\ddot{x}_t \tag{12}$$

Equation (12) indicates that a damping force due to electromagnetic induction occurs in addition to one due to kinetic friction.

To examine the static potential energy characteristics of the vibration model, $\ddot{x} = 0$ and $\dot{x} = 0$ are substituted into Equation (12), which results in the following equation expressing the potential energy:

$$U = Kx^2 - 2Kl_0\sqrt{x^2 + h^2} \tag{13}$$

Next, the distribution characteristics of the potential energy can be examined by differentiating it from Equation (13) to obtain the roots of equation.

$$K\left(1 - \frac{l_0}{\sqrt{x^2 + h^2}}\right)x = 0 \tag{14}$$

For the extremes of the derived potential energy, which yields the following results:

$$x_1 = -\sqrt{l_0^2 - h^2} \ x_2 = 0 \ x_3 = \sqrt{l_0^2 - h^2} \tag{15}$$

Here, $x_1$, $x_2$, and $x_3$ in Equation (15) are the extreme values of the potential energy and correspond to the static equilibrium positions of the mass block. A graphical representation of the potential energy expressed in Equation (13) is presented in Figure 6, where $x_1$, $x_2$, and $x_3$ correspond to the extreme values of the potential energy in Equation (15). $x_1$, and $x_3$ in the valley correspond to the centers of two monostable vibrations on the left and right, respectively, whereas $x_2$ in the peak corresponds to the center of the bi-stable vibrations on the central axis of symmetry. $\Delta U$ represents the barrier value of the bi-stable vibration to the left and right.

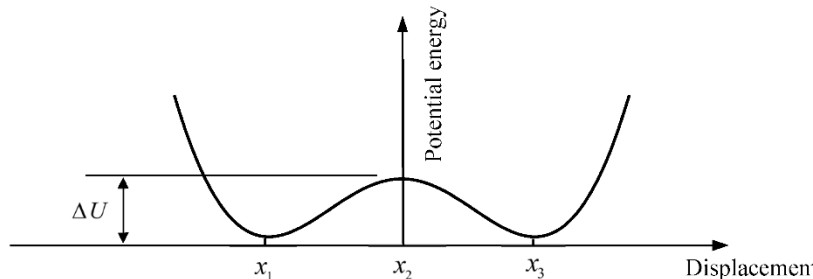

**Figure 6.** The distribution of potential energy with displacement of mass block.

Typically, bi-stable motion models that are placed in a random vibration environment exhibit monostable motion, whereas when they are vibrating simultaneously with random and periodic signals, a large amount of bi-stable vibration occurs, and resonance may be realized, where the vibration displacement is amplified significantly. However, in terms of the uncertain factors that arise in a random excitation environment, this type of resonance phenomenon is referred to as stochastic resonance.

*2.2. Numerical Analysis*

In the equation of vibration for the mass block, i.e., Equation (12), the following symbols are adopted:

$$\zeta = \frac{c}{2\sqrt{mk}} \tag{16}$$

$$\beta = \frac{B^2 n^2 b^2}{mR} \tag{17}$$

$$\omega_n = \sqrt{\frac{K}{m}} \tag{18}$$

Here, $\zeta$ represents the damping ratio, $\beta$ the extent of the damping effect caused by electromagnetic induction, and $\omega_n$ the natural angular frequency of the spring-mass model.

The relative expansion/contraction ratio of the spring is expressed as a function of $L(x)$, which is expressed as.

$$L(x) = \frac{l_0}{\sqrt{x^2 + h^2}} - 1 \tag{19}$$

The equation of vibration for the mass block is.

$$\ddot{x} + [2\xi\omega_n + \beta W(x)]\dot{x} - \omega_n^2 L(x)x = -\ddot{x}_t \tag{20}$$

Subsequently, the equation of vibration above can be rewritten in the form of simultaneous equations, as follows:

$$\frac{d\dot{x}}{dt} = -[2\xi\omega_n + \beta W(x)]\dot{x} + \omega_n^2 L(x)x - \ddot{x}_t \tag{21}$$

$$\frac{dx}{dt} = \dot{x} \tag{22}$$

Because the relative displacement $x_i$ and relative velocity $\dot{x}_i$ of the mass block are derived at each time step using the Runge–Kutta method for Equations (21) and (22), the relative displacement equation, i.e., Equation (9), and the electromagnetic induction equation, i.e., Equation (5) can be employed to obtain the following numerical solution for the vibrational displacement and voltage at each time step.

$$x_{di+1} = x_{i+1} + x_{ti} \tag{23}$$

$$V_{i+1} = BnbW(x_{i+1})\dot{x}_{i+1} \tag{24}$$

### 2.3. Magnetic Flux Density

Using the numerical analysis method described in the previous section, the magnetic flux density $B$ of the magnetic field near the coil must be provided to analyze the vibrational displacement of the mass block and the voltage of the vibrational power generation. In this study, the magnetic flux density $B$ near the coil was measured directly using a Tesla meter (model number WT10A), as illustrated in Figure 7. The $B$ near the coil was measured thrice and averaged, which yielded a value of 70 mT.

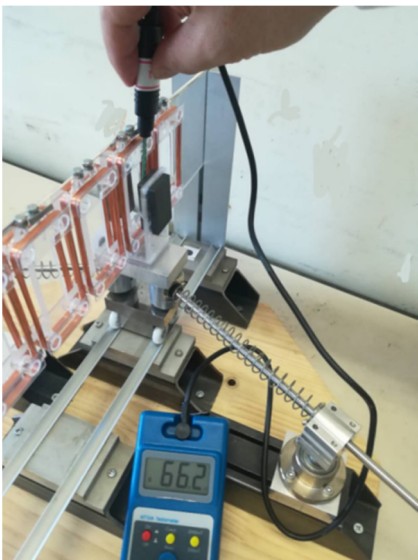

**Figure 7.** Measurement of flux density of magnetic field around coil.

### 2.4. Identifying Damping Coefficients and Confirming Accuracy of Analysis

The damping effect occurring in a vibrating mass block includes the damping force $F_c$ caused by kinetic friction and the Lorentz force $F_L$ yielded by the induced current flowing in the magnetic field. The Lorentz force $F_L$ can be estimated using Equation (11), whereas the damping force $F_c$ is uncertain, and the damping coefficients must be identified in advance. In this study, the damping coefficients were identified by performing the following steps:

(1) The general shaker maintained stationary and measured using only the mini-shaker as the excitation source.

(2) The bi-stable vibration model was vibrated with a sinusoidal wave of frequency 1.5 Hz. Subsequently, the vibration displacement $x_{\exp er}$ and induced voltage $V_{\exp er}$ were measured and recorded under conditions of electromagnetic inductive damping.

(3) A damping coefficient $c$ was assigned as a tentative value.

(4) The vibration displacement $x_i$ and voltage $V_i$ were estimated based on a numerical analysis method derived using the Runge–Kutta method.

(5) The values of $x_i$ and $V_i$ obtained numerically were compared with the values of $x_{\exp er}$ and $V_{\exp er}$ measured experimentally. If the errors are significant, then the damping factor $c$ is adjusted, and the analysis is continued by re-performing step (4) until both errors become adequately small. Finally, a damping coefficient $c$ is obtained.

Here, when adjusting the damping coefficient, there is only one adjustment parameter, and the numerical analysis is a simple calculation performed in Excel. The area where the attenuation coefficient exists is equally divided, and the results of each analysis are compared with the experimental values. The damping coefficient of the case with the smallest error was taken as the identified value.

Using the structural parameters of the experimental model, an identification process was performed for damping coefficient A, which resulted in damping coefficient $c = 2.31$ Ns/m. The corresponding vibration displacements and voltages are shown in Figures 8 and 9, respectively. The blue line shows the numerical values, the dotted red line shows the experimental measurements, and black line shows the error between experimental and analytical values; as shown, although there are some errors during the vibration process, the trends of the two results are well agree with each other. $c = 2.31$ Ns/m is intended to be used as all the damping coefficients of the bi-stable vibration model in this study. Results obtained using the proposed numerical analysis method and the configuration weight function in Equation (8) indicate that the analytical accuracy is adequate.

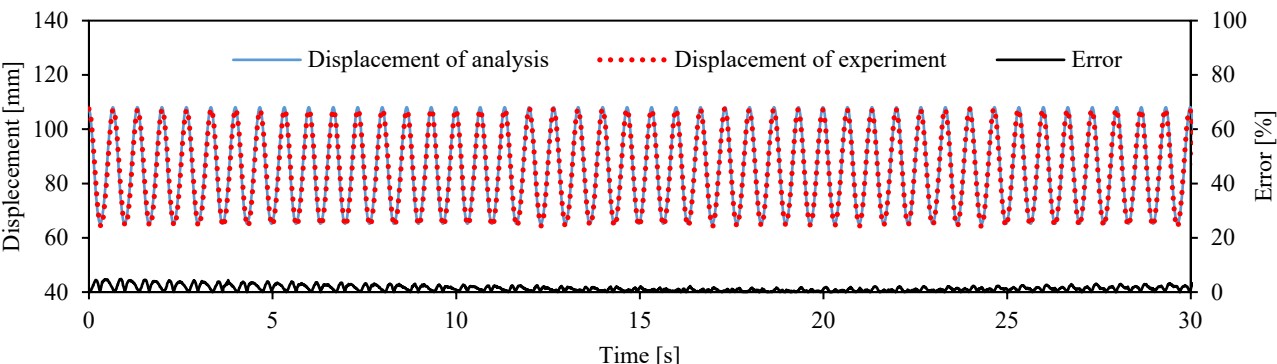

**Figure 8.** Graphs of displacement based numerical analysis vs. time for confirming damping coefficient.

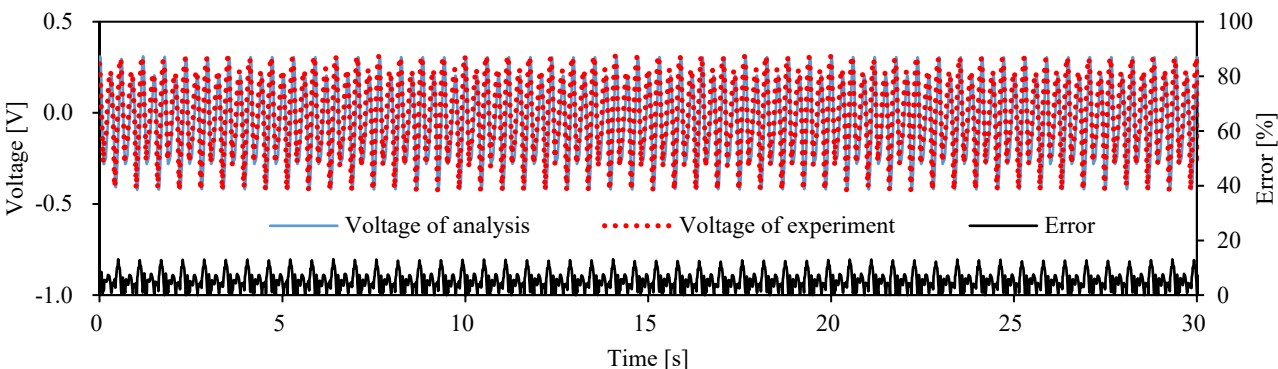

**Figure 9.** Graphs of voltage-based analysis voltage vs. time for confirming damping coefficient.

### 3. Results

To confirm the adequacy of the analytical accuracy, measurement experiments were performed for three experimental cases: (1) random signal excitation, (2) periodic signal excitation, and (3) simultaneous excitation of both excitation signals (i.e., random and periodic signals).

The natural frequencies $f_1$ at the stable vibration points on both sides of the bi-stable vibration system and $f_2$ at the central unstable vibration point are determined using Equations (25) and (26), respectively.

$$f_1 = \frac{1}{2\pi}\sqrt{\frac{\left|U''\left(\sqrt{l_0^2 - h^2}\right)\right|}{m}} = \frac{1}{2\pi}\sqrt{\frac{2K}{m}\left(1 - \frac{h^2}{l_0^2}\right)} \tag{25}$$

$$f_2 = \frac{1}{2\pi}\sqrt{\frac{|U''(0)|}{m}} = \frac{1}{2\pi}\sqrt{\frac{2K}{m}\left(\frac{l_0^2}{h} - 1\right)} \tag{26}$$

Substituting the structural parameters into Equations (25) and (26), the results show that the natural frequencies at the stable vibration points on both sides are $f_1 = 1.65$ Hz, whereas those at the unstable vibration point in the center are $f_2 = 1.33$ Hz. Furthermore, based on the results of preliminary experiments conducted in advance, the frequency of the periodic vibration signal was set from 1.0 to 2.2 Hz at intervals of 0.3 Hz, and the amplitude of the periodic vibration signal was set uniformly at 20 mm.

To quantitatively evaluate the effect of stochastic resonance amplification, the standard deviation $S$ (in mm), which was calculated using the following formula, was applied as an index for evaluating the vibration displacement.

$$S = \sqrt{\frac{1}{N}\sum_{i=1}^{N}(x_i - \overline{x})^2} \tag{27}$$

Here, $x_i$ is the measured value of the vibration displacement, $\overline{x}$ the average value of the vibration displacement, and $N$ the number of discrete vibration displacement values in the time series.

The efficiency of the vibration power generation is estimated using the power quantity $P$ (in mW), which is calculated as follows:

$$P = \frac{1}{T}\int\frac{V^2}{R}dt = \frac{1}{TR}\sum_{i=1}^{N}V_{\exp er-i}^2\Delta\Delta t \tag{28}$$

Here, $R$ represents the electrical resistance, $V_{\exp er-i}$ the measured voltage at each sampling measurement point in the vibration experiment, $N$ the total number of steps in the total measurement time, $\Delta t$ the step time width, and $T$ the total measurement time.

### 3.1. Measurement Results Yielded by Random Signal Excitation

The results yielded by the random signal excitation were measured and are shown in Figure 10. The black line represents the vibration displacement of the mass block, the blue line indicates the vibration displacement of the support point, and the red line indicates the voltage of the vibration power generation.

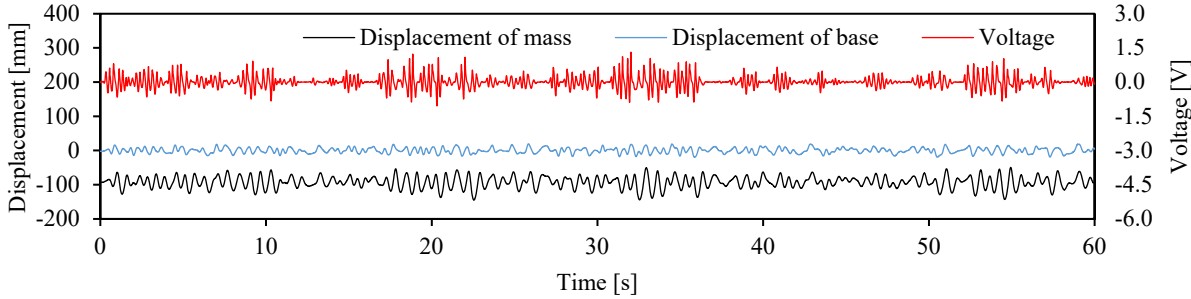

**Figure 10.** Measurement results yielded by random signal excitation.

Based on Figure 10, the vibration displacement was relatively small when a random signal excitation was applied, and the mass block featured a single stable vibration on the right side of the central axis of symmetry because the vibration originated from the right side of the mass block. The standard deviation of the vibration displacement of the mass block was 17.22 mm, and that of the support point was 7.92 mm, thus resulting in a vibration power of 1.99 mW.

### 3.2. Measurement Results Yielded by Periodic Signal Excitation

As shown in Figures 11–15, owing to the periodic signal excitation, the vibration displacements exhibited the same periodicity, whereas the voltage of the vibration power exhibited a little different periodic distribution as it was affected by the frame shape of the coil.

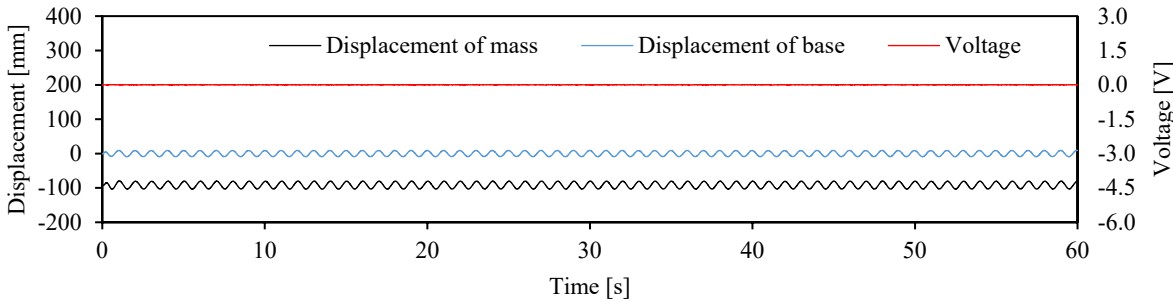

**Figure 11.** Measurement results for 1.0 Hz periodic signal excitation.

No relative displacement was observed between the mass block and support point, as they both vibrated periodically simultaneously and did not generate power when vibrating at a low frequency such as 1.0 Hz, as shown in Figure 11. The standard deviations of the vibration displacements of the mass block and support point were 7.93 and 6.35 mm, respectively, which resulted in 0.00 mW power generated.

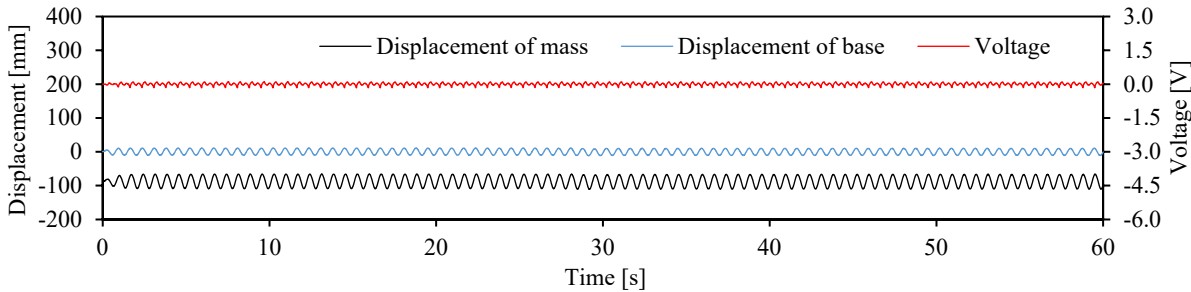

**Figure 12.** Measurement results for 1.3 Hz periodic signal excitation.

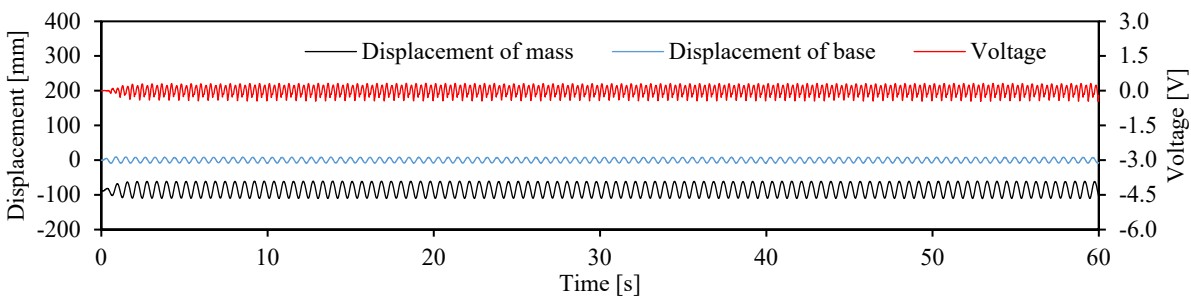

**Figure 13.** Measurement results for 1.6 Hz periodic signal excitation.

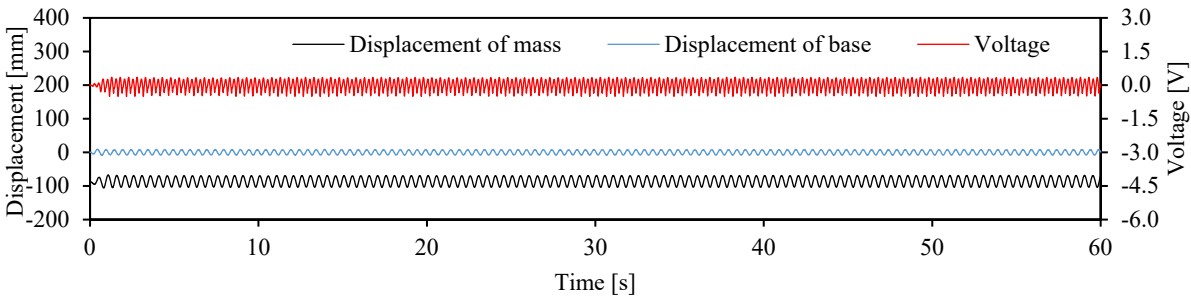

**Figure 14.** Measurement results for 1.9 Hz periodic signal excitation.

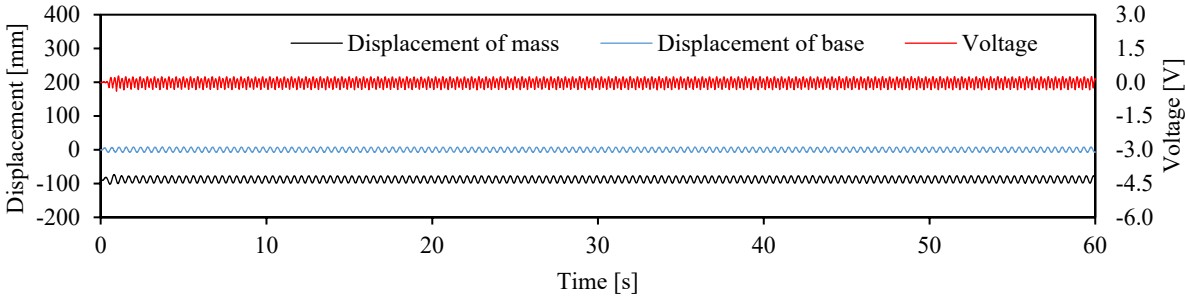

**Figure 15.** Measurement results for 2.2 Hz periodic signal excitation.

When the excitation frequency was set to 1.3 Hz, as illustrated in Figure 12, the mass block vibrated periodically with an amplitude greater than that of the support point. The standard deviations of the vibration displacement of the mass block and support point were 14.94 and 7.39 mm, respectively, which resulted in 0.12 mW of power generated.

The amplitude of the mass block increased further and exceeded that of the support point when the excitation frequency was 1.6 Hz, as shown in Figure 13, which might be because the excitation frequency was similar to the natural frequency of the vibration model. The standard deviations of the vibration displacement of the mass block and support point were 16.87 and 5.71 mm, respectively, which resulted in 1.28 mW of power generated.

When the excitation frequency was 1.9 Hz, as shown in Figure 14, the vibration of the mass block was less intense than that at the excitation frequency was 1.6 Hz. The standard deviations of the vibration displacement of the mass block and support point were 12.03 and 5.62 mm, respectively, which resulted in 1.63 mW of power generated.

When the excitation frequency was 2.2 Hz, as shown in Figure 15, the amplitude of the mass block was smaller than that at a lower excitation frequency, such as 1.0 Hz. The standard deviations of the vibration displacement of the mass block and support point were 7.34 and 5.33 mm, respectively, which resulted in 0.80 mW of power generated.

The effects of the periodic signal frequency on the vibrational displacement and the amount of vibrational power generated are summarized in Figures 16 and 17, respectively. In Figure 16, the red and blue bar graphs show the vibration displacements of the mass block and support point, respectively. As the excitation frequency increased, the vibration displacement of the support point increased significantly, whereas that of the mass block peaked at approximately 1.6 Hz.

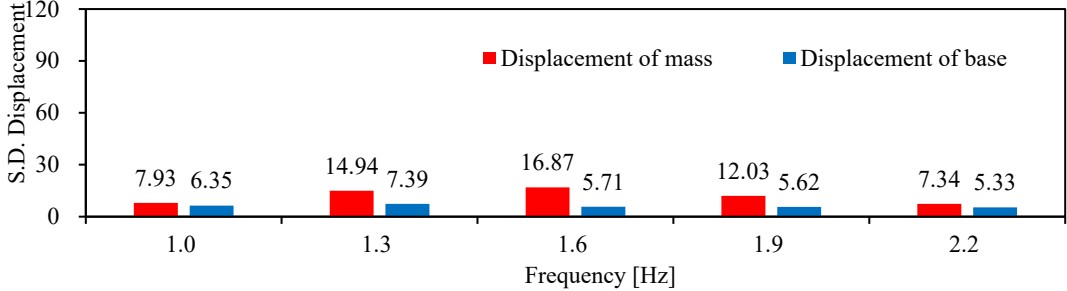

**Figure 16.** Effect of periodic excitation signal frequency on vibration displacement.

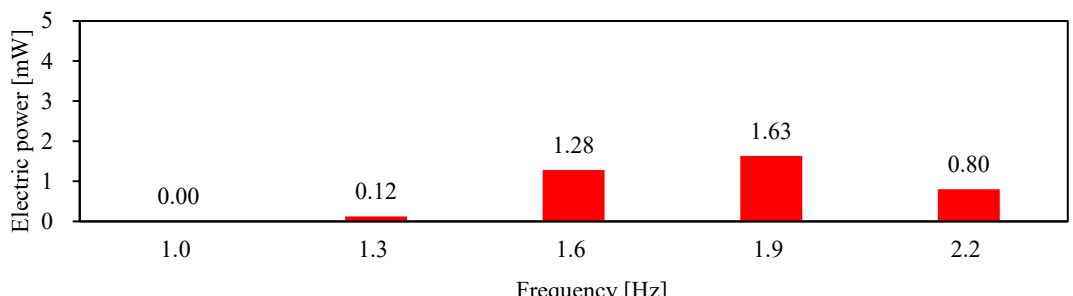

**Figure 17.** Effect of periodic excitation signal frequency on vibration power generated.

Based on Figure 17, one might assume that the amount of vibration power generated increases with the frequency. However, the amount of vibration power generated reached its maximum at approximately 1.9 Hz, whereas at 1.0 Hz, the amount of vibration power generated was zero as no relative displacement was indicated between the mass block and support point.

### 3.3. Measurement Results Yielded by Random and Periodic Signal Co-Excitation

The results yielded by the joint excitation of the random and periodic signals are shown in Figures 18–22. At the periodic signal of 1.0 Hz (see Figure 18), the vibration displacement of the mass block slightly exceeded that of the support point, which resulted in a single stable motion with standard deviations of 21.45 and 11.76 mm for the vibration displacements of the mass block and support point, respectively; consequently, 1.04 mW of power was output.

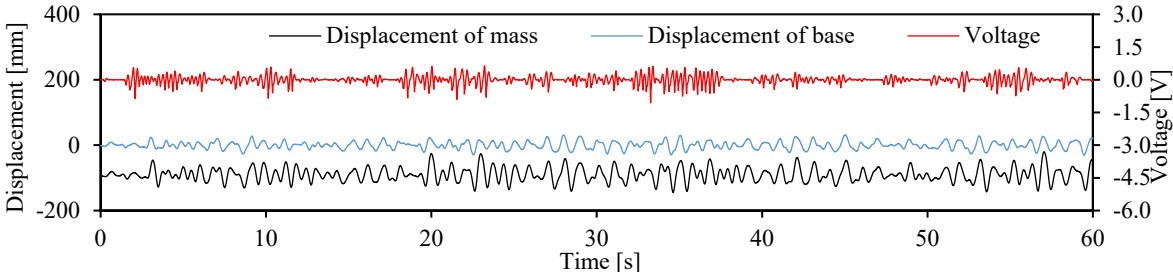

**Figure 18.** Results based on co-excitation of random and 1.0 Hz periodic signals.

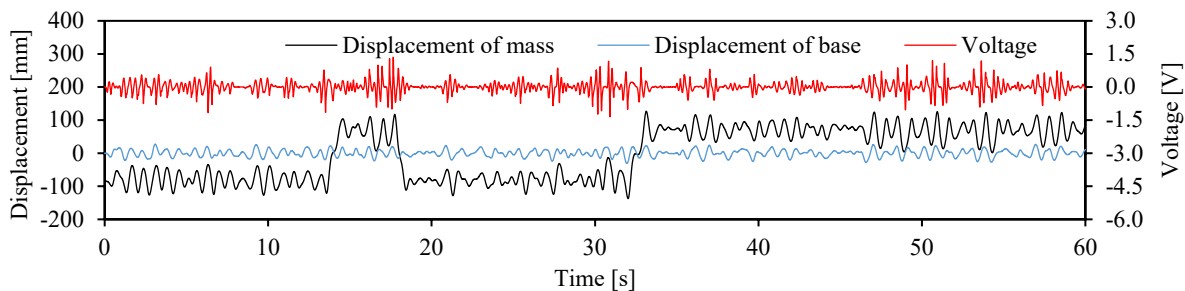

**Figure 19.** Results based on co-excitation of random and 1.3 Hz periodic signals.

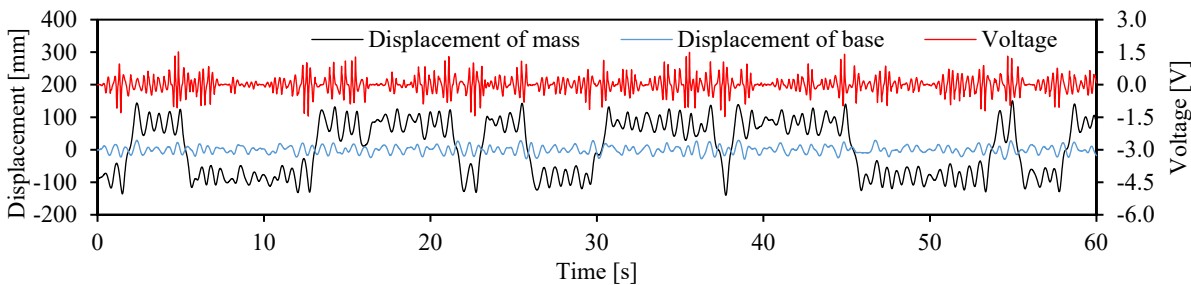

**Figure 20.** Results based on co-excitation of random and 1.6 Hz periodic signals.

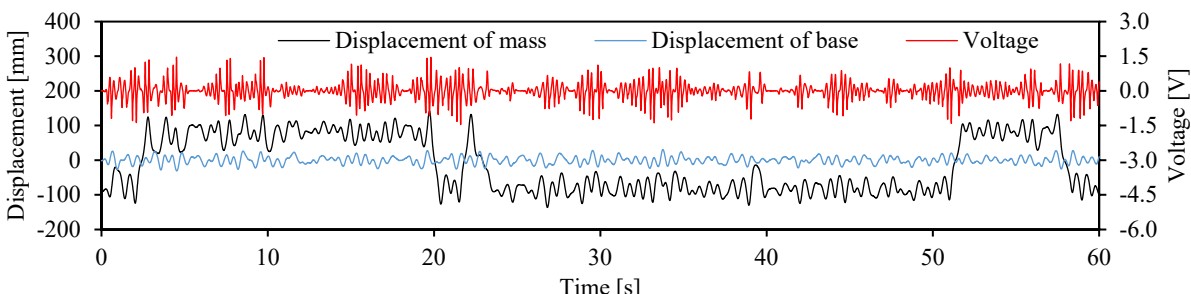

**Figure 21.** Results based on co-excitation of random and 1.9 Hz periodic signals.

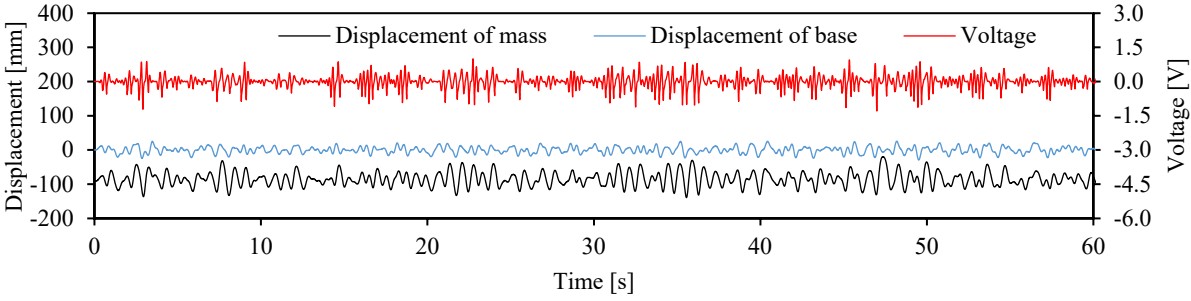

**Figure 22.** Results based on co-excitation of random and 2.2 Hz periodic signals.

When the periodic signal frequency was 1.3 Hz, as shown in Figure 19, the vibration of the mass block became more intense, and a stochastic resonance phenomenon occurred that caused the monostable vibration of the mass block to evolve to a bi-stable vibration with standard deviations of 77.64 and 10.85 mm in the vibration displacements of the mass block and support point, respectively; consequently, 2.08 mW of power was output.

Meanwhile, at a higher periodic signal frequency of 1.6 Hz, the mass block vibrated more intensely, as shown in Figure 20, and the number of occurrences of bi-stable vibration increased, indicating that stochastic resonance is most likely to occur in this excitation frequency range. The standard deviations of the vibration displacement of the mass block and support point were 83.47 and 11.53 mm, respectively, which resulted in 3.51 mW of power generated.

The number of occurrences of bi-stable vibration in the mass block began decreasing as the periodic signal frequency approached 1.9 Hz, as presented in Figure 21, with standard deviations of 80.74 and 10.92 mm for the vibration displacements of the mass block and support point, respectively, which resulted in 3.76 mW of power generated. When the periodic signal frequency shown in Figure 22 approached 2.2 Hz, the vibration of the mass block reduced, the bi-stable vibrations disappeared, and the mass block indicated monostable vibrations. The standard deviations of the vibration displacement of the mass block and support point were 20.70 and 9.84 mm, respectively, which resulted in 2.18 mW of power generated.

The effects of periodic signal frequencies on the vibration displacement and vibration power generated are summarized in Figures 23 and 24, respectively. Based on Figure 23, the vibration displacement of the support point did not change substantially as the excitation frequency increased, but the vibration displacement of the mass block amplified significantly from 1.3 to 1.9 Hz owing to the stochastic resonance associated with the bi-stable vibration. The increase rate of vibration displacement was 615.50% at 1.3 Hz, 623.94% at 1.6 Hz and 639.38% at 1.9 Hz, respectively.

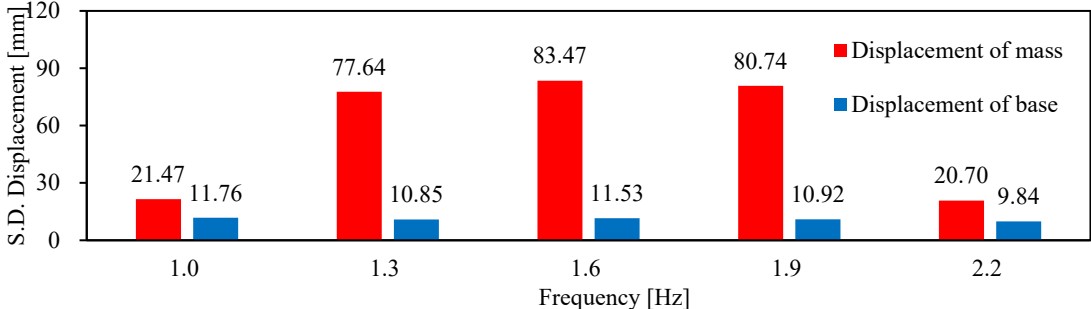

**Figure 23.** Comparison of S.D. displacements of random and period excitation.

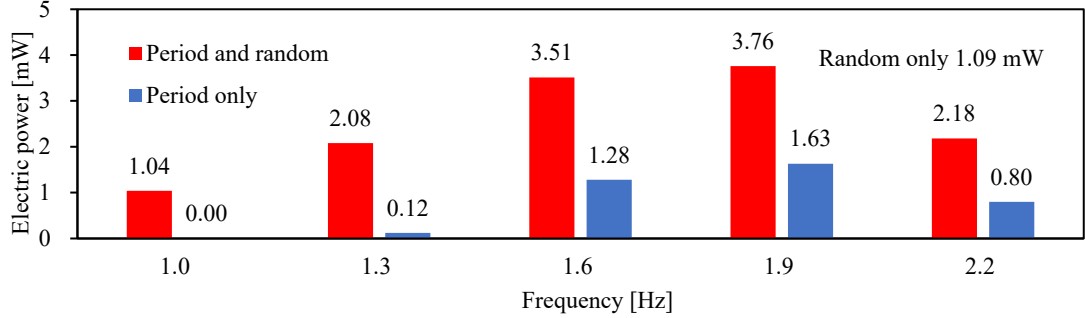

**Figure 24.** Effect of periodic excitation frequency on power output.

Based on Figure 24, the highest vibration power was generated from 1.3 to 1.9 Hz, owing to the mutual effect of the increase in vibration frequency and the stochastic reso-

nance. The rate of increase in power generation was 71.90% at 1.3 Hz, 48.10% at 1.6 Hz, and 38.24% at 1.9 Hz, respectively.

## 4. Discussion

### 4.1. Effect of Electromagnetic Induction Damping on Vibration Displacement

To examine the effect of the electromagnetic induction damping newly generated by the Lorentz force produced when the current derived from the vibration power generated flows in the coil in the magnetic field on the vibration of the mass block, two experimental cases, as illustrated in Figure 25, were established for a comparative study. (1) An ordinary magnet, illustrated on the left in Figure 25, was affixed to the surface of the mass block to measure the vibration resulting from the electromagnetic induction damping effect. (2) The magnet was removed, as illustrated in the right figure of Figure 25, and a clay of the same weight was adhered onto the top of the mass block to measure the vibration resulting from the absence of the electromagnetic induction damping effect.

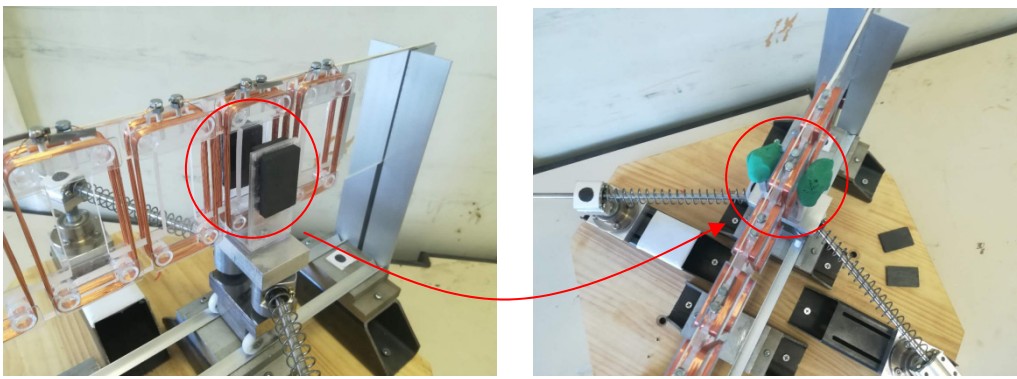

**Figure 25.** Displacement measurement with and without magnet.

For the two experimental cases under the same conditions except for the electromagnetic induction damping effect, the measurements obtained are shown in Figure 26. The blue and red lines indicate the vibration displacements resulting from the presence and absence of the electromagnetic induction damping effect, respectively. Figure 26 shows that the effect of electromagnetic inductive damping on the vibration response displacement is minimal, and that the overall vibration amplitude of the mass block is marginally lower when the electromagnetic inductive damping effect is considered.

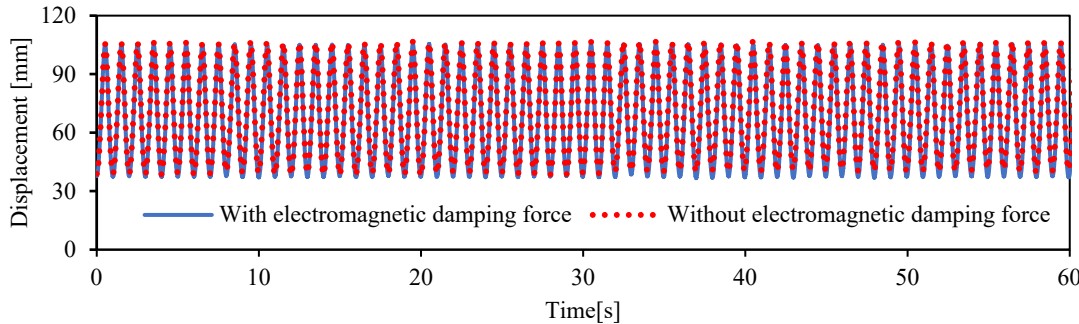

**Figure 26.** Effect of electromagnetic damping on vibration displacement.

This distinction above can be investigated quantitatively by comparing the coefficients of the damping force in Equation (12). The equivalent damping coefficient $c_L$ resulting from the electromagnetic induction for the measured damping coefficient $c = 2.15$ is calculated as follows:

$$c_L = \frac{B^2 n^2 b^2}{R} W(x) \tag{29}$$

By substituting the detailed parameters of the bi-stable vibration model in Table 1 into Equation (29) and considering the configuration weighting factor $|W(x)| \leq 1$, an equivalent damping coefficient resulting from electromagnetic induction is obtained as follows:

$$c_L \leq 0.0199 \tag{30}$$

Therefore, the damping force resulting from electromagnetic induction is less than 2% of the total damping force.

*4.2. Forecasting the Most Likely Frequency at Which Stochastic Resonance Occurs*

The periodic excitation frequency at which the stochastic resonance is the most likely to occur can be determined using Kramer's rate, as follows [15]:

$$f_k = \frac{\omega_1 \omega_2}{4\pi q} \exp\left(-\frac{\Delta \overline{U}}{D}\right) \tag{31}$$

Here, $\omega_1$ refers to the equivalent natural angular frequency at the stable equilibrium point $x_1 = \pm\sqrt{l_0^2 - h^2}$, $\omega_2$ the equivalent natural angular frequency at the unstable equilibrium point $x_2 = 0$, $\Delta\overline{U}$ the barrier value of potential energy calculated using Equation (20), $q$ the damping factor including electromagnetic induction damping effects, and $D$ the random signal intensity. $\omega_1$, $\omega_2$, $\Delta\overline{U}$, and $q$ are expressed as follows:

$$\omega_1 = \sqrt{\frac{\left|U''\left(\sqrt{l_0^2 - h^2}\right)\right|}{m}} \tag{32}$$

$$\omega_2 = \sqrt{\frac{|U''(0)|}{m}} \tag{33}$$

$$\Delta\overline{U} = \frac{1}{m}\left[U(0) - U\left(\sqrt{l_0^2 - h^2}\right)\right] = \frac{K}{m}(l_0 - h)^2 \tag{34}$$

$$q = \frac{1}{m}\left(c + \frac{B^2 n^2 b^2}{2R}\right) \tag{35}$$

The random signal intensity $D$ was calculated using the time-series response displacement of the support table, which was measured when only the random signal alone was excited. The calculated $D$ was 0.319 J/kg.

Substituting Equations (32)–(34) and $D$ into Equation (31), the periodic signal frequency at which stochastic resonance tends to be generated can be expressed as follows:

$$f_k = \frac{K(l_0 - h)}{2\pi q l_0}\sqrt{\frac{l_0 + h}{h}} \exp\left(-\frac{K(l_0 - h)^2}{mD}\right) \tag{36}$$

Substituting the structural parameters into Equation (36) yields $f_k$ = 1.64 Hz as the frequency at which stochastic resonance is the most likely to occur, which is within the frequency range of 1.3–1.9 Hz discussed in the previous section.

## 5. Conclusions

A horizontal bi-stable vibration energy-harvesting system was proposed herein. The conclusions inferred from investigating the system are as follows:

(1)  An energy-harvesting system was proposed for application to random vibration environments by employing a vibration power generation unit comprising magnets and coils. By establishing a set of governing equations that simultaneously consider the elastic force of the spring and the Lorentz force of electromagnetic induction, the potential energy performance of the proposed system was analyzed, and the results

showed that the system exhibited bi-stable vibrational characteristics over the entire range of motion. An arrangement weight function $W(x)$ as proposed that incorporated the mutual positional relationship between the magnet and coil during the vibration process, thus enabling a quantitative analysis of the voltage generated in the coil. The analytical values of vibration displacement and voltage derived from numerical analysis were consistent with the measured experimental values.

(2) To determine the friction damping and electromagnetic induction damping of the bi-stable vibration model, a damping coefficient identification method combining numerical analysis and experimental measurements was employed to analyze the actual damping coefficients. The results yielded showed agreement with the experimental values. The average error of the vibration displacement was 2.46%, and the average error of the voltage was 5.27%. To quantitatively investigate the effect of electromagnetic induction damping on the vibration power generated, the experiment results showed that the electromagnetic induction damping force was about 2% smaller than the normal friction damping force.

(3) The appropriate frequency range of periodic signals were added to generate stochastic resonance. Subsequently, by investigating with Kramer's rate, a prediction equation for the frequency range of periodic signals within which stochastic resonance can be generated easily was derived, and its validity was verified based on comparison with the results of experimental measurements. When random and periodic signals were excited simultaneously, the proposed bi-stable vibration energy-harvesting system effectively improved the vibration amplification and vibration power generation performance by ensuring the generation of stochastic resonance. Its average vibration displacement increased by 629.29%, and the average power generation increased by 52.75%.

**Author Contributions:** Writing—original draft preparation, L.G.; writing—review and editing, J.G.; data curation, X.Z.; investigation, N.G.; software, W.Z.; conceptualization, L.G. and X.Z.; methodology, J.G.; validation, W.Z. and N.G. All authors have read and agreed to the published version of the manuscript.

**Funding:** This research received no external funding.

**Data Availability Statement:** Data available in a publicly accessible repository.

**Conflicts of Interest:** The authors declare no conflict of interest.

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
