# Peer review of "Horizontal Bi-Stable Vibration Energy Harvesting Using Electromagnetic Induction and Power Generation Efficiency Improvement via Stochastic Resonance"

_machines, doi:10.3390/machines10100899_

Round 1

Reviewer 1 Report

(1)The main problem in this paper is that the main point is not prominent, and the innovation is not obvious. The stochastic resonance is used to improve the efficiency of the vibration power generation. Compared with the existing practices, what are the advantages of the method adopted in this paper?

(2) This paper points out that ‘When random and periodic signals were excited simultaneously, the proposed bi-stable vibration energy-harvesting system effectively improved the vibration amplification and vibration power generation performance.’ Please, explain on the experimental setup how to excite both stochastic and periodic signals.

(3) Figures 8 and 9 suggest that error curves be presented and interpreted. Figure 26 suggests to adjust the line shape.

(4) The conclusion is slightly redundant. It is suggested to integrate into three points, and it is better to have data support in each conclusion.

(5) The English expression of this article needs to be significantly strengthened. And the format should also be optimized.

Reviewer 2 Report

     In this paper, a horizontal bistable vibration energy harvesting system based on elastic spring and mass block is proposed, and a set of governing equations considering elastic force and electromagnetic induction Lorentz force of elastic spring are derived. The factors affecting the output capacity of the harvester, including damping coefficient and external excitation source, are analyzed in detail. The whole paper is innovative and suggested to be accepted after some modification.

(1)    I think Table 1 should be adjusted to a three-line table.

(2)    In Section 2.1, the legend of Figure 4 is different from the description of Figure 4 above in the paper. Please check and modify it carefully.

(3)    A total of 55 references are listed at the end of the paper. However, references [56] and [58] appear in the paper. Please carefully check the references in the whole paper.

(4)    In Section 2.4, the identification process of damping coefficient is mainly described. The vibration displacement x of the harvester is compared with the experimental numerical results of the induced voltage V, and the error is adjusted to take the damping coefficient in a smaller range. However, how to calculate the error and adjust the damping coefficient is the key for identification, and the authors should explain this process in detail.

(5)    In Section 3, the author tests the output capacity of the vibration harvester under different external excitations, including its vibration displacement and output voltage. I think that the vibration displacement and output voltage should be simulated and compared with the experimental results to further verify the accuracy of the proposed mathematical model.

Reviewer 3 Report

In this paper, the Authors write that the Horizontal bi-stable vibration energy harvesting using electromagnetic induction and power generation efficiency improvement via stochastic resonance is created.

In the beginning, I would like to write that article is interesting, especially the test stand, but there is a lot to consider or change to improve the overall clarity of this work.

My remarks:

-          fig. 1 - nice but too low quality

-          fig. 1, 2 – there is no indication on the picture(-s) about shaker vibration direction

-          fig. 3 – “Vibration device”

-          there is no information about recording video parameters

-          table 1. Elastic spring – is the spring coefficient and initial length calculated to be working as a part of a quasi-zero-stiffness system?

-          overall formatting of the article is weak, too small symbols in the text and in equations, mistakes in describing it, a lot of white spots, and typos (fig. 3, fig. 8 (displacement, also a legend – displacement of analysis? or “based on numerical analysis”, fig. 5 – Lc? Or Lc), line 294 – V (italic), and so on …

-          fig. 9 – voltage over the entire coil system or individual coils?

-          eq. 25, 26, 32, 33 – derivative notation not consistent (Lagrange notation, Leibnitz notation, Newton notation)

-          fig. 4 – I do not understand what the double red arrow means. There are no symbols, no information about that

-          on every image the thickness of the dimension lines is incorrect, I know that this is not a strictly technical drawing, but changing them will improve the readability of the drawings

-          fig. 5 – is the waveform in the figure between -3, -2.5 and 2.5, 3 a drawing error? or a deliberate procedure? (Irregular sine wave shape)

-          maybe I missed it, but I do not see functions describing the work of vibration inductors and their implementation in the equations, the test results contain only measurements on a real object, they were not related to the results obtained using a numerical model, please explain whether the model was to determine certain parameters or to develop it further, what is the purpose of the numerical model shown. I also think that it would be good for the future to present the model in a dimensionless form, which will significantly reduce the number of parameters and speed up the calculations.

Round 2

Reviewer 1 Report

(1) Please systematically revise the abstract and introduction to highlight the main point, especially the introduction;

(2) The error curves in Figures 8 and 9, please explain in the text;

(3) The English expression part needs to be strengthened;

(4)The format should also be optimized, such as the format of the references should be unified.

Reviewer 2 Report

  • The author has corrected some questions ,so I suggest publishing.

Author Response

Thank you for your valuable comments on our manuscript.